# Perception Bias Effects on Healthcare Management in COVID-19 Pandemic: An Application of Cumulative Prospect Theory

**DOI:** 10.3390/healthcare10020226

**Published:** 2022-01-25

**Authors:** Tienhua Wu

**Affiliations:** Department of Management, Air Force Institute of Technology, Kaohsiung 82047, Taiwan; 9428901@nkust.edu.tw

**Keywords:** coronavirus disease 2019 (COVID-19), healthcare decisions, perception bias, attitude, long-term care facilities (LTCFs), cumulative prospect theory (CPT)

## Abstract

Coronavirus disease 2019 (COVID-19) has posed severe threats to human safety in the healthcare sector, particularly in residents in long-term care facilities (LTCFs) at a higher risk of morbidity and mortality. This study aims to draw on cumulative prospect theory (CPT) to develop a decision model to explore LTCF administrators’ risk perceptions and management decisions toward this pandemic. This study employed the policy Delphi method and survey data to examine managers’ perceptions and attitudes and explore the effects of sociodemographic characteristics on healthcare decisions. The findings show that participants exhibited risk aversion for small losses but became risk-neutral when considering devastating damages. LTCF managers exhibited perception bias that led to over- and under-estimation of the occurrence of infection risk. The contextual determinants, including LTCF type, scale, and strategy, simultaneously affect leaders’ risk perception toward consequences and probabilities. Specifically, cost-leadership facilities behave in a loss-averse way, whereas hybrid-strategy LTCFs appear biased in measuring probabilities. This study is the first research that proposes a CPT model to predict administrators’ risk perception under varying mixed gain–loss circumstances involving considerations of healthcare and society in the pandemic context. This study extends the application of CPT into organizational-level decisions. The results highlight that managers counteract their perception bias and subjective estimation to avoid inappropriate decisions in healthcare operations and risk governance for a future health emergency.

## 1. Introduction

Identified as a pandemic by the World Health Organization (WHO) in January 2020, coronavirus disease 2019 (COVID-19) remarkably affected the healthcare sector and rapidly evolved into a global emergency [1,2]. This disease is harmful to all populations, particularly individuals residing in long-term care facilities (LTCFs) who require specialized medical care and life support [2,3]. LTCF residents are a vulnerable population group at a higher risk of susceptibility and mortality from COVID-19 than younger population groups [4,5,6]. The high morbidity and mortality of COVID-19 presents challenges for surveillance systems and infection prevention and control (IPC) and causes severe damage to safety, healthcare provision, and administrative operations in LTCFs, leading to higher reported deaths compared to younger population groups [7,8].

Since this public health emergency was identified, governments and institutions have released guidelines or implemented regulations to meet the demands of LTCFs while mitigating the impacts of the crisis [1,5]. However, considerable health, psychology, and socioeconomic damage highlight the existence of managerial problems at the system and facility levels [4,9,10,11]. The European Centre for Disease Prevention and Control (ECDC) [7] suggests that the lack of unique surveillance systems, differences in testing strategies and capacities, and LTCF staff’s abilities are the main factors contributing to facility-wide outbreaks. Renda and Castro [10] note that the reasons underlying this emergency are insufficient investments and fragmented efforts in governing low-risk, high-consequence events such as COVID-19. D’Adamo et al. [4] also suggest that governments underscore coronavirus disease risk spreading among elderly LTCF populations. Healthcare leaders’ responses to the threat of COVID-19 remain weak and differ across facilities.

This outbreak should have been predicted and avoided given the sufficient experience and knowledge learned from SARS or H1N1; however, it was underestimated or even ignored by decision-makers in the COVID-19 context [10,11]. The decisions for mitigating the impacts of the pandemic on healthcare operations may include risk assessment and response [12], medical supplies and inventory management [13], or comprehensive preparedness and contingency plans [14]. Notably, since decision-makers are risk-sensitive, their attitudes affect strategy selection and implementation [15]. Liu et al. [16] argue that emergency choice problems are complicated because of the evolution of disaster scenarios and the uncertain information. Indeed, the rapidly changing situations make clinical and care management decisions complex [4,17]. If managers have a bias that prevents them from making decisions effectively and efficiently, devastating health and economic outcomes may result [4].

The abovementioned discussion emphasizes human factors in making decisions related to care provision and risk practices. The extant literature inadequately addresses this issue. To bridge this research gap, the following research questions arise:

RQ *1*: How do administrators in LTCFs define and evaluate unprecedented adverse events such as COVID-19?

RQ *2*: How do administrators’ cognitive heuristics and risk-related behaviors influence their decisions in response to the pandemic?

RQ *3*: If administrators have biases, what factors contribute to these biases, and how can these biases be adjusted to ensure effective decision-making under risk?

This study uses infection risk in the ongoing COVID-19 pandemic affecting healthcare operations in LTCFs as the research setting. To respond to the research questions, this study first conducts the policy Delphi method of data collection to explore varying LTCF administrators’ risk perceptions and attitudes regarding COVID-19 and contextual factors affecting healthcare management. Then, this study draws on cumulative prospect theory (CPT) to build a model examining how risk perceptions and attitudes of managers impacted their decision in detecting, evaluating, and managing risk during the outbreak. CPT has been widely applied in studying individuals’ behaviors and choices under various risk contexts [18,19]. Last, based on policy Delphi results and a literature review [20,21,22,23,24,25], this study identifies and incorporates individual and organizational variables in the proposed model to examine bias effects on healthcare management decisions by the survey data. The results contribute to the literature by proposing a CPT decision-making model to capture managers’ risky choice behaviors when facing an unprecedented outbreak. The present study also helps practitioners examine bias effects of individual and organizational factors on decisions, better plan proper countermeasures in healthcare management decisions, and reorient risk practices in the post-COVID-19 period.

This study proceeds as follows: Section 2 provides a literature review on the impacts of the pandemic on LTCFs, CPT, and factors affecting healthcare decisions. Section 3 presents the methods and decision tasks, a CPT-based decision model, and a regression model for sociodemographic factors. This study then provides the results regarding administrators’ perceptions and attitudes toward infection risk and the distortive effects of individual and organizational factors on managers’ decisions. This study discusses the results in Section 5 and summarizes the implications in Section 6.

## 2. Literature Review

### 2.1. Research on the Impact of COVID-19 and Policy Delphi Method

The literature highlights that COVID-19 has dramatically impacted long-term care facilities (LTCFs), a specific group of facilities in the healthcare sector [1,7]. LTCFs care for people with physical, mental, intellectual, or sensory disabilities who require specialized medical care and life support. Given physical or mental disabilities and a high probability of infection clusters, residents are particularly vulnerable populations who suffer the worst consequences from this infection [1,6,7]. Evidence reports that the deaths of residents in many EU countries correspond to 30~60% of the total COVID-19 deaths [7,26], and half of the infected people in LTCFs in the United States are asymptomatic, making it difficult to estimate the numbers of residents and staff affected by COVID-19 [27]. Such situations may result from underestimating the coronavirus risk [4,8] or failures at the systems level [9].

In response, managers and staff require additional preparedness in terms of workforce capacity, resources, care planning, and organizational efforts to reduce safety damage while sustaining healthcare operations [1,7,28]. However, these requirements post many decision nodes that require accurate responses by healthcare personnel to achieve transmission mitigation. For example, D’Adamo et al. [4] suggest that managers and staff require awareness of ongoing situations, appropriate managerial behaviors, containment strategies, and decisions on all levels of governance. Leite et al. [29] emphasize the importance of dealing with the operational dilemma of capacity and demand when the infection curve is generated. Bhaskar et al. [13] and Kuo et al. [14] stress managers’ efforts and responsiveness of organizations in governing their medication supply and contingency plans to reduce damages. Therefore, the current pandemic has exposed the vital roles of human factors and managers’ risky behaviors in clinical and managerial decisions in the current and post-COVID-19 period [7,12].

This indicates a need for healthcare personnel to generate viewpoints on contextual factors affecting decisions under COVID-19 risk. In the healthcare setting, the literature highlights the benefit of the policy Delphi method for generating ideas through structured, critical collective debate within a group of anonymous experts [30,31,32]. For example, Paraskevas and Saunders [33] highly recommend using the policy Delphi to generate options regarding organizational crisis signals detection and suggest alternative courses of action for consideration. Meskell et al. [32] suggest affordance of the policy Delphi method in nursing research, which identifies best practices in the clinical role of nurse lecturers by examining stakeholders’ perceptions. Despite the weaknesses such as inconsistent execution process and incomplete data [30,31], De Loë et al. [30] note that the policy Delphi method is well suited for inquiry into complex problem areas with multiple perspectives solutions with no one clear normative solution. Thus, this study employed the policy Delphi method to understand LTCF managers’ perceptions of this unprecedented risk.

### 2.2. Cumulative Prospect Theory and Its Applications

To explain variations that cannot be predicted by rational decision theory, Kahneman and Tversky [34] and Tversky and Kahneman [35] proposed prospect theory (PT) and CPT. PT and CPT used the value function (VF) and the probability weighting function (PWF) to observe individuals’ risky choice behaviors such as reference dependence, loss aversion, diminishing sensitivity, and probability weighting. These behaviors may result in irrationality, cognitive bias, and nonlinear transformation in a decision, leading to a situation in which risks may not be appropriately predicted, evaluated or managed [19]. In CPT, the VF relates to the subject’s irrational valuation of money, i.e., the nonlinear perception of the action monetary spending. The function in VF is asymmetric and exhibits risk aversion behavior if it is steeper for significant losses than it is for small losses. The PWF measures the likelihood of event occurrence on the perceived money prospects. The PWF carries the nonlinear transformation from objective to subjective probabilities for each individual [18,19,34,35,36]. Individuals, constrained to cognitive limitations, are usually challenged to rationalize the fair value of probabilistic events. Thus, biased individuals tend to overweight low probabilities and underweight high probabilities.

CPT has been widely applied in a variety of contexts, including economics and business [18,37,38,39], public emergencies and catastrophic events [16,40], and the healthcare domain [41,42,43]. The goals of all of the studies above are to further study human agents’ risk-related behaviors in uncertain environments and their decisions regarding risky choices, which help managers adjust their choices to manage risks [16,39]. Given the importance of agents’ risk-related behavior in COVID-19 control, this research extends prior work by proposing a CPT decision-making model to capture managers’ risky choice behaviors when facing this unprecedented outbreak.

### 2.3. Factors Affecting Healthcare Decisions

Scholars [20,21,22,23,24,25] have highlighted the significance of individual and organizational variables in framing, assessing, and managing risks. The literature addresses the vital roles of individual factors such as personal characteristics and expert knowledge elicitation [20], domain experience [21], or gender equity [22] in decisions under risk. Concerning organizational variables, March and Shapira [23] suggest that managers’ decisions are particularly affected by their attention to critical performance targets. McNamara and Bromiley [24] find that factors such as organizational standardization or pressure for profitability appeared to influence executives’ assessment of risky decisions. Additionally, Cagliano et al. [20] note that errors may have roots grounded in people highly influenced by the working environment and the organizational processes.

During the COVID-19 pandemic, studies have shown that human errors have contributed to disease spread within and between LTCFs. For example, policymakers and healthcare personnel failed to identify outbreaks earlier [7,10], attended to infections with limited IPC training and equipment [7,26,27], and underestimated the transmissibility rate and disease burden [4,7]. Leite et al. [29] acknowledge that healthcare organizations must face varying difficulties in managing COVID-19 demand, such as lack of system thinking in decision-making or staff issues regarding numbers and safety concerns. Variations in personnel’s risk perception are the primary sources that result in their different risk-taking behavior in response to infection risk [21]. Thus, the individual difference in healthcare decisions under risk warrant additional research in the context of this unexpected, severe pandemic.

While the WHO and CDC (Centers for Disease Control and Prevention) provide IPC guiding principles for LTCF stakeholders, they do not consider differences in organizational features and capacities, the health conditions of residents, the staff’s abilities, or their resource supply models. Actual measures and actions may depend on different regions and facilities [44]. In the hospital setting, the extant literature identifies that organizational variables, such as strategy types [45,46,47] or business environments and contexts [48,49], affect managerial decisions and performance. Similarly, organizational characteristics and strategy types of LTCFs should be considered in managers’ decision-making, given the complex political, legal, societal, business, and financial environments created by COVID-19. Based on the abovementioned discussion, this study thus focuses on six factors: gender, job title, funding status, LTCF type, facility scale, and strategy in influencing decision-making that benefit better planning of proper clinical and managerial countermeasures in the light of future health emergencies.

## 3. Methodology

This study employed a multi-methods approach. First, a two-round policy Delphi technique was used to explore information about risk management faced by LTCFs during the COVID-19 pandemic and design decision tasks for the CPT decision model. Second, drawing on CPT, this study proposed a decision model to examine managers’ perceptions and attitudes toward pandemics and a regression model to explore the effects of sociodemographic characteristics on healthcare decisions. Last, the survey method was utilized to examine the models above to gain advanced knowledge of human biased effects on decision making and provide countermeasure suggestions.

### 3.1. Decision Task Design and Factors Using Policy Delphi Method

The policy Delphi method uses iterative stages of exploring the broadest range of insights or revealing possible opposing positions on an issue rather than generating consensus among anonymous panelists [30,50], and it is a reliable means because of its four primary characteristics: anonymity, iteration with controlled feedback of group opinion, statistical aggregation of group response, and expert input [51]. Meskell et al. [32] suggest that the policy Delphi technique is highly recommended in a healthcare setting because healthcare organizations are hierarchically structural, and junior personnel tend to avoid challenging their superiors’ opinions. Moreover, the policy Delphi technique aims to draw out the different views and is a tool for analyzing complex issues [50]. The COVID-19 emergency is a complicated issue in which evaluation and responses to this adverse event vary among various LTCF stakeholders in vastly different IPC conditions and disrupt resource supply and logistics. Therefore, in the COVID-19 context, the policy Delphi is a suitable method for understanding varying LTCF administrators’ risk perceptions to design decision tasks to examine individual and organizational factors in influencing managers’ healthcare decisions.

The policy Delphi is a multiphase process. This method includes six phases: formulation of the issues, exposing the options, determining initial positions on the issues, exploring the obtaining of the reasons for disagreements, evaluating the underlying reasons, and re-evaluation the options [50]. The first three phases deal primarily with statements, arguments, and comments on an issue from varying perspectives of participants [32,33,50]. The fourth and fifth phases focus on assessment and justifications, and the last phase is to re-evaluate the alternatives with acceptability or feasibility [32,33,50].

This study used two Delphi rounds conducted from June to November 2020. Following the research of De Loë et al. [30] and Meskell et al. [32], interviewees were recruited by two principles: whether interviewees were knowledgeable about healthcare operations and supply strategies and had decision-making authority with pandemic risk. Interviewees were informed to (1) generate ideas about risk management for COVID-19 and (2) act as representatives of their facilities rather than as individual anonymous interviewees. After explaining the details associated with research aims and procedures, seven interviewees were recruited, and all maintained participation in the subsequent rounds. Data were collected using face-to-face interviews, providing opportunities for LTCF representatives to express the different views of the COVID-19 pandemic from their facilities’ perspectives [51].

### 3.2. CPT Decision Model and Measurement of Prospect Values

PT and CPT suggest that in a choice process, individuals first edit the available options as prospects that are a representation of outcomes and probabilities, evaluate the edited prospects, and eventually choose the one with the highest value as jointly determined by the VF and the PWF [34,35]. The VF relates a subject’s nonlinear perception of monetary spending. This CPT model used VFs to describe subjects’ valuation of money on infection cluster losses derived from the entrance of possible infected or asymptomatic agents [35]. The PWF measures the likelihood of event occurrence on perceived monetary prospects. Probabilities represented the risk associated with the entrance of possibly infected or asymptomatic stakeholders. Participants’ attitudes toward these probabilities were shown by CPT’s PWFs [35].

Complementing the straightforward expected utility theory, CPT incorporates human bias into utility functions v(⋅) and w(⋅), as shown in (1). The model parameters are estimated by an intermediate measure of certainty equivalent (CE):(1)v(CE)=w(p)v(X)+(1−w(p))v(Y)

The procedure involves collecting individuals’ responses and performing nonlinear regression on the curves of v(⋅) and w(⋅). The functional forms are employed according to the suggestion of [35]:(2)$$v\left(X\right)=-\lambda {\left(-X\right)}^{\alpha },\;\mathrm{for}\;X<0,\;\mathrm{and},$$
(3)$$w\left(p\right)=p^{\beta }/{\left(p^{\beta }+{\left(1-p\right)}^{\beta }\right)}^{1/\beta }.$$

To estimate model parameters α and β in (2) and (3), this study solved nonlinear regression (1) by the Markov Chain Monte Carlo (MCMC) method. The MCMC technique has the advantage of numerical stability, statistical accuracy, and computational efficiency in handling significant problems. This study used the MCMC method to estimate posterior probabilities of the target parameters α and β in a solver of OpenBugs 3.2.3, a publicly available MCMC software. In contrast to previous studies in iterative approximating approaches [16,35,52,53], this study used the nonlinear fitting and obtained the posterior probability p(α,β|CE,v,w) based on likelihood functions (p(v|α),p(w|β)). In nonlinear regression, it is unnecessary to assume Gaussian distribution for any error terms. 

This study rewrote (1) into an indexed form (4) with corresponding sampling sets. For individual i, j-th loss probability $p_{j}$, and k-th loss value $X_{k}$, this study fitted the model with an additional error term ϵ in CPT Equation (4): (4)$$v\left(CE_{ijk}\right)=w\left(p_{j}\right)v\left(X_{k}\right)+w\left(1-p_{j}\right)v\left(0\right)+\epsilon .$$

Equation (4) can be explained as a respondent facing situations in decision Tasks. The first loss value is USD 10,000 $\left(X_{1}=10000\right)$, and the first loss probability is 1% $\left(p_{1}=0.01\right)$. Because the choice of investment is either $X_{1}$ or nothing, the alternative to $X_{1}$ is 0. The CE value is USD 1000 $\left(CE_{111}=1000\right)$. This study tentatively assumed the error term to be Gaussian, and it turned out to be well represented in the simulation of the MCMC iterations.

This study also explored the roles of individual and organizational variables in influencing agents’ risky choice behaviors. To address this research aim, this study used a regression analysis technique involving six sociodemographic variables: gender, job title, the funding status of the organization, LTCF type, facility scale, and organizational strategy type. This study estimated the model as follows and used $\theta_{1}\sim \theta_{6}$ to represent the six factors above, respectively.
(5)$$v\left(X\right)=-\lambda {\left(-X\right)}^{\alpha },w\left(p\right)=p^{\beta }/{\left(p^{\beta }+{\left(1-p\right)}^{\beta }\right)}^{1/\beta },$$
(6)$$\alpha =\gamma_{0}+\overset{6}{\underset{i=1}{\sum }}\gamma_{i}\theta_{i}+\eta ,$$
(7)$$\beta =\xi_{0}+\overset{6}{\underset{i=1}{\sum }}\xi_{i}\theta_{i}+\zeta ,$$
where γ, ξ are the regression coefficients and η, ζ are the regression errors. The α parameter values for the factor coefficients represent the effects of individual and organizational variables on the VFs of respondents, whereas the values of the β parameter show factor effects on the PWFs.

### 3.3. Survey Method

The survey was conducted from February to April 2021. This study first contacted the potential long-term care organizations in Taiwan via telephone to ask whether they were willing to participate in this study. Next, this study mailed paper questionnaires regarding organization strategy and decision tasks in the Appendix A to participating facilities and asked administrators to respond to the survey. Concerning strategy, participants were asked to indicate on a seven-point scale the extent to which, compared with other LTCFs, their care services (1) provided innovative, differentiated, diverse, and large-scale services/programs and (2) engaged in a cost-efficient analysis of equipment and resources, facilities, workforce, and services and programs. This study received the questionnaires by mails, and there were no personal identification data on the paper questionnaires. A total of 327 questionnaires as a sample size were used to examine the CPT decision models. According to the definition of LTCF types by ECDC [44], participants’ facilities were three types: general nursing homes, residential homes, and mixed LTCFs. Table 1 provides the sample characteristics. Respondents who gave a higher sum value on engaging in innovative services were the differentiation group, comprising approximately 43.8% of the participating facilities. Participants who indicated a higher sum value on performing cost control were assigned to the cost-leadership group; 25% of all respondents were from cost-leadership organizations. Approximately 31.2% of organizations were hybrid organizations whose sum values were between cost-leadership and differentiation groups.

## 4. Results

### 4.1. Policy Delphi Results

The first round was designed to incorporate phases 1–3 of the design suggested by Turoff and Linstone [50]: formulation of issues, exposing the options, and determining initial positions on the issues. In the first round, this study reviewed the relevant literature: CPT, the study of Guo, et al. [39], and IPC guidance released by the WHO and ECDC for questions and preliminary IPC decision tasks. Interviewees provided their views on pandemic perceptions, contextual factors, critical agents, and occurrences and impacts of this virus transmission. Interviewees agreed that possible infected agents’ movement and health conditions, such as residents, staff, and visitors, should be screened daily to effectively identify new infections and prevent further spread [1,4]. They also suggested that pre-symptomatic or asymptomatic cases are hard to identify but remarkably contribute to disease transmission [26,27]. Similar to situations in Europe [44], interviewees emphasized re-assessing workplace risk under occupational safety and health legislation and therefore need to find a way to minimize the workload for facilities while achieving an acceptable cost–benefit ratio. Moreover, they pointed out that an additional burden for environmental cleaning, waste management, and daily reporting depends on LTCF types and scale and challenges administrators and healthcare staff. Except for mandatory IPC regulations, interviewees’ actions to IPC practices widely differed, and organizational factors could determine the extent to which a facility engages in healthcare operations to control COVID-19 risk. Overall, data revealed the facility’s weakness to IPC practices and the gaps regarding workforce and resource capacities and demand [29].

Interviewees in the second round aimed to gain an advanced understanding of views that emerged from round one [32,51] and helped modify preliminary IPC decision tasks for a survey purpose. This round focused on exploring and obtaining the reasons for disagreements and evaluating the underlying reasons suggested by Turoff and Linstone [50]. Due to high transmissibility rates, possibly infected and asymptomatic stakeholders were identified and monitored as these cases had varying possibilities to enter facilities and cause different degrees of impact. Interviewees noted that infection control would add a significant burden that might be difficult to fulfill without additional financial and medical resource provision or affecting primary care. They thus modified monetary amounts of decision tasks representing facilities’ investment in IPC practices and the consequences of confirmed cases and infection clusters. Meanwhile, interviewees also modified the risk probabilities regarding the entrance of possibly infected or asymptomatic stakeholders. To evaluate the effects of facilities’ strategies, interviewees agreed to adopt the measurements from the study of Kumar and Subramanian [46] to identify the extent to which LTCFs use cost, differentiation, or hybrid strategies to provide healthcare services.

The decision task questionnaire was refined in the second round then mailed to survey respondents to rank preference on decision tasks as phase 6 of Turoff and Linstone [50]. The questionnaire has two task circumstances, in which possibly infected or asymptomatic stakeholders enter facilities, leading to varying degrees of losses represented by monetary values. To avoid losses, managers make a risky choice by transforming the representation of the entrance occurrence probabilities with loss outcomes.

Taking Task 1 as an example, the entrance probability of a possibly infected stakeholder is assumed to increase from 1% (i.e., one possible agent among 100 agents who entered facilities) to 3%, 10%, 30%, and then 90%. The entrance of possible cases lead to losses in the amounts of USD 10,000, 40,000, 70,000, 100,000, or 130,000 (Scenarios 1.1~1.5, respectively). In response, numerous choices regarding investment in IPC activities, ranging from USD 1000, 3000, 10,000, 30,000, or 100,000, are offered. In each loss scenario of Task 1, administrators’ decisions depend on how they edit numerous possibilities and losses under risks and evaluate and select the preferred investment choices in an essential, continuing manner. Similarly, each manager is asked to complete decision Task 2 where the entrance occurrence of asymptomatic stakeholders with different possibilities (i.e., from 0.1% up to 9%) leads to tremendous financial losses, ranging from USD 0.2 to 1.4 million (Scenarios 2.1~2.5). The amount of IPC investment includes five choices: USD 10,000, 30,000, 100,000, 300,000, or 1,000,000. The manager is also asked to assess and choose the acceptable amounts of IPC investments under different probabilities for each loss scenario 2.1~2.5 of Task 2.

Figure 1 shows the scenarios of the decision tasks in the questionnaire. The upward one depicts scenario 1.1 of Task 1, and the downward one is for scenario 2.1 of Task 2. Scenarios 1.1~1.5 are the same except that the loss amount increases from USD 40,000, 70,000, 100,000, to 130,000. Similarly, the loss scenarios of 2.2~2.5 are the same except the questionnaire indicates various loss amounts: USD 500,000, 800,000, 1,100,000, or 1,400,000. Every participant makes decisions in five scenarios of each task based on their perceived probabilities and impacts of outcomes. Given the cognitive limitation and varying perceptions of COVID-19, managers’ risky choices could be observed and modeled in a CPT-based decision model.

### 4.2. Estimation Results of Risk Perception and Attitude

The survey data were used to examine the two CPT functions: the VF and the PWF. The technique is a nonlinear fitting procedure. For each decision task, participants edited choice problems and selected their preferred 25 CEs for 25 different combinations of possibilities and loss outcomes. First, this study examined the validity of a CPT model, separately fitted the function parameters for each task using the MCMC method, performed 320,000 iterations in the MCMC procedure, and discarded the data of the first 6000 iterations. As shown in Figure 2, the probabilities of the parameters such as λ, α, β, and ϵ approximately fit a Gaussian distribution. Table 2 exhibits the results of the nonlinear fitting. The mean of the error term ϵ in the CPT equation is $8.657\times 10^{-7}$; all Monte Carlo errors are less than 0.025, suggesting that the proposed CPT model is acceptable.

Second, this study examined the α values shown in Table 2, representing agents’ risk attitudes toward outcomes through the VF. The α value (1.433) for Task 1 is higher than 1. In the context of the entrance occurrence of possibly infected stakeholders, the findings suggest that most respondents exhibit risk-averse behavior. Similarly, the estimated α (1.176) for Task 2 is over one but close to 1, indicating that the participants have low risk aversion under conditions in which asymptomatic stakeholders entered into facilities.

Lastly, the β values obtained in this study are used to describe subjects’ attitudes toward probabilities. A greater β means that the risk preference of an agent is more neutral than biased. In contrast, a smaller β suggests that agents present severely biased behaviors to enhance small probabilities and reduce higher probabilities. As shown in Table 2, β values for Tasks 1 and 2, 0.525 and 0.382, respectively, are smaller than 1, implying that respondents’ probability judgmental distortions under the circumstances of Tasks 1 and 2.

### 4.3. Regression Results for the Effects of Individual and Organizational Factors

Table 3 shows the regression results in which some coefficients of the factors are significantly different from 0, suggesting distortive effects of individual and organizational factors exist. Regarding VFs, the greater the α value is, the more risk-averse a participant behaves. Sociodemographic factors, including LTCF type, facility scale, and strategy type, significantly affect participants’ VFs. The coefficient values of α parameter row of −0.06, 0.1, and −0.03 suggest that administrators of general nursing homes, larger facilities, and cost-leadership organizations, respectively, have cognitive value bias. Particularly, large-scale organizations with a coefficient value of 0.1 exhibit higher risk-averse behaviors.

The values of the β parameter row show factor effects on the PWFs. The results in Table 3 identified all six variables’ bias effects on the perceived likelihood of an event occurrence. Given that our coding assigned values of 1 for females and 2 for males, the 0.09 coefficient of gender suggests that male executives are generally more sensitive to changes in probabilities than their female counterparts. Compared to medical administrators, facility managers with a coefficient value of −0.16 tend to distort infection risk probabilities or decision weights more. The estimated coefficient value of −0.27 suggests that public LTCF managers are more likely to enhance low probabilities and reduce high probabilities than for-profit organizations. Similar to the VFs, LTCF factors such as type, scale, and strategy, particularly in the general nursing homes, larger LTCFs, and hybrid care organizations, affect participants’ PWFs, leading to biased behaviors when estimating the occurrence of infected cases.

## 5. Discussion

### 5.1. Risk Perception and Attitude

Table 2 shows that VFs of participants exhibit different risky behavior when facing varying decision tasks. The α value for Task 1 (1.433) is higher than that for Task 2 (1.176). The graphed results in Figure 3 show that the curves on the top left and bottom left panes are slightly concave for infection risk of possible cases (i.e., Task 1) and linear for infection risk of asymptomatic stakeholders (i.e., Task 2). Administrators tend to invest relatively large amounts in compensating for comparatively small losses resulting from possible infection cases, but they become risk-neutral or even risk-seeking when considering relatively damaging losses caused by this pandemic. In line with the previous literature [15,19,35], our results suggest that executives’ diminishing sensitivity to the VFs for increasingly damaging losses thus determines how facilities engaged in healthcare preparedness and medical supply respond to COVID-19.

Regarding PWFs, Table 2 shows that administrators underweight large probabilities and overweight small probabilities in both tasks, as predicted by CPT [19,35]. LTCF administrators show severe biased weighting behavior in which they may have inadequate awareness of emergencies such as COVID-19 and may not make optimal decisions to prevent the possible adverse effects. Task 2 has a smaller β value (0.382) than Task 1 (0.525). The right column results of Figure 3 show that more considerable bias in probability weighting exists when managers weigh asymptomatic stakeholders’ entrance occurrence (Task 2) than that of possibly infected cases (Task 1). The findings also suggest that the PWFs vary depending on agents’ attitudes toward different risk circumstances, such as the entrance occurrence of possible or asymptomatic cases. Overall, the results support that managers’ risk perception and behavioral characteristics can affect decisions because of human bounded rationality in disaster response [16,25,52].

This study further considered the VF and the PWF jointly, suggesting risk attitude as a combination of attitudes toward outcomes and probabilities [19]. In Task 1 of possibly infected cases, as the PWF result of Task 1 distorts less significantly than Task 2 (see the right column of Figure 3), managers’ decisions are more influenced by their risk-averse behavior response to this pandemic. By contrast, as the VF for Task 2 is close to linear, managers’ risk response is entirely determined by the PWF rather than jointly by the functions v(⋅) and w(⋅). The findings highlight that decision-making can be influenced by either probability weighting or outcome sensitivity or a combination of probability and outcomes, leading to over- or under-estimation of risks and choice of inappropriate strategies. Hence, practitioners need to consider human cognitive limitations in risk preference and management. If agent bias could be counteracted, it is possible to avoid the same healthcare mistakes during the outbreak [10,12], reduce bias impacts on optimal decisions for medical resource governance [13,14], or raise awareness of inadequate contingency preparedness and risk response for healthcare operations [29]. Overall, our evidence fits the CPT elements proposed by Kahneman and Tversky [34] and Tversky and Kahneman [35], reflecting healthcare leaders’ loss aversion and bias in response to the ongoing COVID-19 pandemic.

### 5.2. Biased Effects of Individual and Organizational Factors

Consistent with the findings of early studies on risk-taking behaviors [20,21,23,24,25], our results empirically identify subjective effects of either individual or contextual factors or both on editing and evaluating outcomes of complex choice situations. Regarding VFs, although the influence of individual variables is not significant, our findings identify the distortive effects of organizational factors, i.e., LTCF type, facility scale, and strategic orientation, on investment valuation for IPC practices [23]. In particular, large-scale organizations with a more considerable *α* coefficient value of 0.1 exhibit higher risk-averse behaviors and burden leaders to consider monetary spending to avoid relatively more minor losses. To respond to COVID-19, care organizations need to manage varying clinical and non-clinical resources and equipment supported by different sourcing systems and strategies. Thus, the findings highlight the vital roles of organizational characteristics in shifting decision-making toward a broader consideration of facilities’ systems [29].

As for the results of PWFs, all six variables’ biased effects are ratified. Administrators’ risk perceptions that depend on a combination of individual characteristics, mental operations, and decision circumstances may impact their decisions [21,36]. Assessing is exacerbated by the contextual unpredictability and fragmented information involved in agents’ subjective valuation [4,16,36] and their incomprehensive lens on impending business disruptions [11] in this evolving emergency. As shown in Table 3, a public organization, denoted by a higher coefficient value of −0.27, more strongly affects managers’ assessments of probabilities of adverse events than other contextual determinants [10]. Accordingly, personal and organizational aspects require enhanced attention if care organizations want improvements in preparedness and responsiveness for medicine’s long-term quality and security during a pandemic [14].

The regression results for the *α* parameter and *β* parameter rows in Table 3 suggest that the factors of facility type, scale, and strategy simultaneously affect participants’ VFs and PWFs. For considering facility type and scale, administrators of general nursing homes and larger facilities present perception bias in assessing outcomes and probabilities. Except for daily care planning, residents at general nursing homes who need skilled nursing care are at heightened risk of exposure to infection; larger LTCFs tend to enhance surveillance practices to minimize the danger of cluster infections [6,44]. In this regard, this unprecedented emergency makes decisions complicated, suggesting that managers’ reference points vary considerably as their choice situations evolve. For general nursing homes and larger facilities, a participant’s attitude toward risk is thus easily biased [36,41].

Specifically, the results in Table 3 show that the effect of facility strategy on the VF and the PWF is different. Cost-leadership LTCFs with a coefficient value of −0.03 exhibit biased valuation of IPC monetary spending, whereas hybrid-oriented organizations (the coefficient value of 0.14) appear biased in measuring probabilities. The literature notes that cost leaders in hospitals mainly focus on cost control and the efficiency of existing operations [46,47]. Similarly, our findings show that administrators of cost-oriented organizations think about the consequences of COVID-19 risk in a loss-averse way. Further, Kumar and Subramanian [46] and Ghiasi et al. [48] identified that hybrid organizations in the healthcare sector lack a clear focus on strategy formulation and implementation and therefore struggle to sustain their competitiveness and financial performance. Perception bias in PWFs of hybrid LTCF has managerial problems. Over- and underweighting risk probabilities may result in biased decisions, affecting IPC effectiveness and medicine supply responsiveness [4,13].

Overall, bias may occur and be exacerbated by individuals’ subjective valuation of the severity of outcomes and rating of risk occurrence. Unwanted results are likely to happen in the subsequent decision-making process. If cognitive limitations on human reasoning and organizational features are addressed, risky choice problems can be evaluated and managed effectively and efficiently. The present results stress the importance of considering the effects of personal- and organization-specific variables, particularly facility type, scale, and strategy, on estimating risk probability and outcomes when making actual choices to respond to future pandemics.

## 6. Conclusions and Implications

Coronavirus disease 2019 (COVID-19) has posed severe threats to human safety and healthcare continuality and quality, particularly in residents in LTCFs at a higher risk of morbidity and mortality. This study adopted the policy Delphi technique to explore LTCF administrators’ risk perceptions and attitudes, developed a CPT-based decision analysis model to understand their risky behaviors, and finally explored the roles of sociodemographic variables in influencing decisions under risk. The findings show that participants appeared risk-averse in the context of small losses and became risk-neutral when considering extensive, devastating damages. LTCF managers exhibited perception bias that led to over- and under-estimation of the occurrence of infection risk. Participants in general nursing homes, larger care organizations, and cost-leadership LTCFs demonstrated bias concerning this pandemic’s outcomes. All six contextual determinants distort decision weights under risk. Public care organizations present relatively severe distortive weighting behavior. Specifically, organizational factors, including LTCF type, scale, and strategy, simultaneously affect leaders’ risk perception toward consequences and probabilities. Concerning the effect of organization strategy, cost-leadership facilities behave in a loss-averse way, whereas hybrid-strategy LTCFs appear biased in measuring probabilities.

This study provides implications for academia and practice. First, this study extends CPT to risk management in the healthcare sector at an organizational level rather than the individual level, as the extant literature does [42,43]. The findings complement COVID-19 literature by exploring managers’ interpretations (i.e., perception and attitude) on varying mixed gain–loss circumstances involving healthcare demand and capacity considerations that influence healthcare management and risk practices [4,15,29]. Second, this study observes participants’ diminishing sensitivity to consequences and biased probability weighting behaviors. Administrators need to counteract individual subjective estimation of occurrence and outcomes, which likely avoid the human errors and systems failures that this coronavirus has revealed [10,12], make optimal decisions for supply risk governance [13], or enhance contingency preparedness for timely, adequate risk response [14]. Last, given rapidly evolving situations and inadequate information, this study suggests that administrators pay attention to their changing perception and reference points depending on the current state of COVID-19 [41,43] and the affordance and capabilities of their care organizations [27,29]. Specifically, our results suggest that proper risk management strongly depends on organizational characteristics such as LTCF type, scale, and strategy. The issues associated with organizational-level risk assessment and decision behavior should be considered when urging LTCFs to provide continuous care while implementing effective IPC practice and risk management.

Some limitations need to be addressed. First, this study primarily collected data from Taiwanese LTCFs. This obstacle may limit the generalizability of the results to other countries or healthcare organizations such as hospitals. Another limitation is the focus on infection risk, which precludes examining the impacts of other clinical and non-clinical risks and neglects intertwined relationships among other health systems’ adverse events. Future research could consider more decision tasks that complement risky choice situations with other risks or unexpected events such as earthquakes. Last, the literature emphasizes competitiveness and facility intra-relationships [48,54] and complex stakeholders and relationships in the healthcare sector [13,14]. Future studies could consider other sociodemographic factors in examining decision-making under risk and influencing reorienting healthcare strategy, particularly organization-specific considerations when facing ongoing COVID-19 and future pandemics. 

## Figures and Tables

**Figure 1 healthcare-10-00226-f001:**
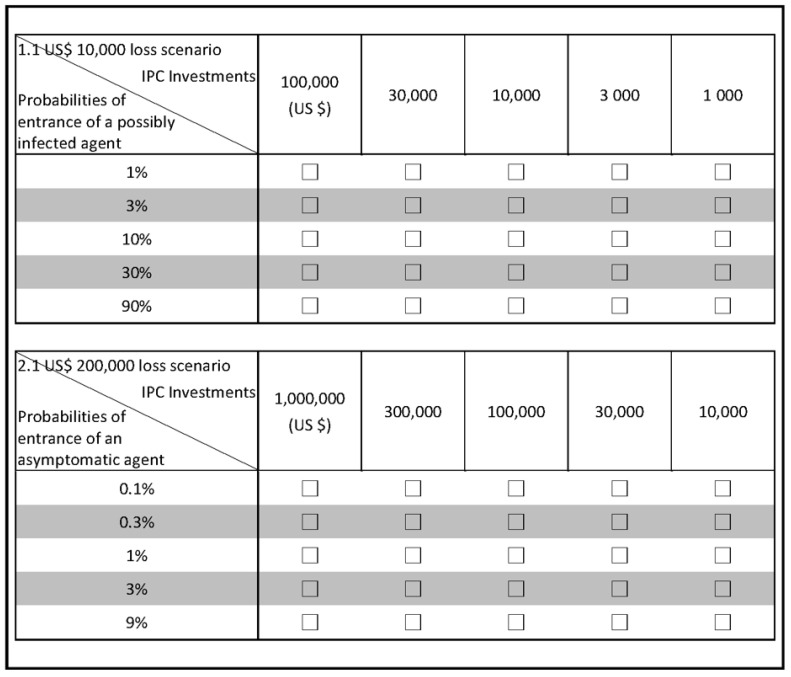
Decision Tasks 1 and 2 under various loss scenarios. The upward one is for scenario 1.1 of Task 1, and the downward one is for scenario 2.1 of Task 2, IPC: infection prevention and control.

**Figure 2 healthcare-10-00226-f002:**
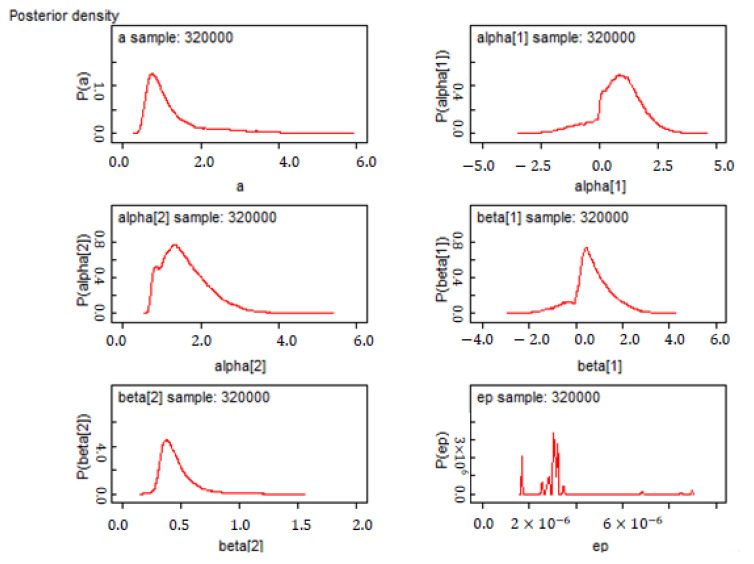
Posterior density distributions for all estimated parameters.

**Figure 3 healthcare-10-00226-f003:**
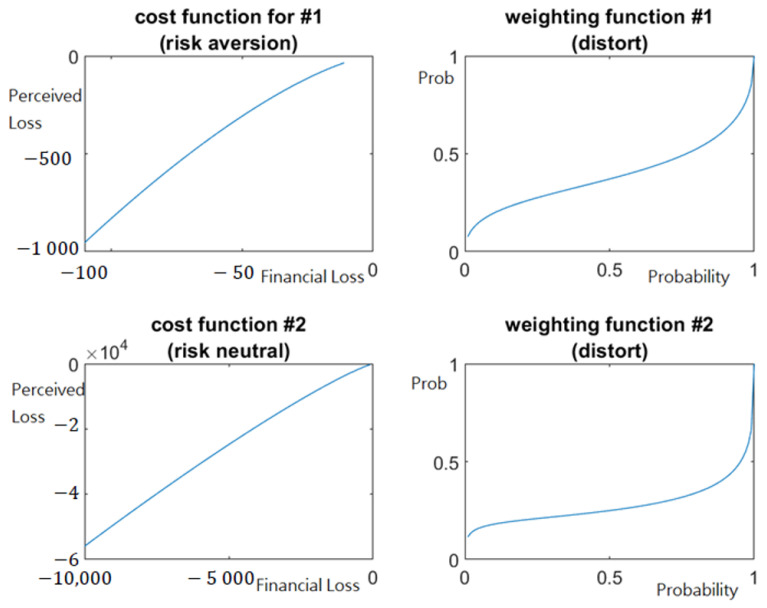
Value function v(⋅) (**left column**) and weighting function w(⋅) (**right column**) for Tasks 1 (**upward row**) and 2 (**downward row**), respectively.

**Table 1 healthcare-10-00226-t001:** Demographics of the participants.

Variables	Items	Frequency	Percent
Gender	Male	123	37.6
	Female	204	62.4
Job title	Facility administrator	102	31.2
	Healthcare/Medical administrator	225	68.8
Funding status	Public	61	18.7
	For-profit	266	81.3
LTCF type	General nursing homes	61	18.7
	Residential homes	102	31.2
	Mixed LTCFs	164	50.1
Facility scale	Less than 99 beds	82	25.1
	100~399 beds	164	50.1
	More than 400 beds	81	24.8

Note: LTCF: long-term care facilities.

**Table 2 healthcare-10-00226-t002:** Results of the nonlinear fitting for Tasks 1 and 2.

Estimated Parameter	Mean	Standard Deviation	MC_Error	Val2.5pc	Median	Val97.5pc	Start	Sample
λ	1.091	0.336	0.010	0.510	1.255	1.388	6001	320,000
α[Task 1]	1.433	0.802	0.024	0.937	0.980	2.879	6001	320,000
α[Task 2]	1.176	0.361	0.014	0.933	0.977	1.816	6001	320,000
β[Task 1]	0.525	0.265	0.008	0.363	0.375	0.988	6001	320,000
β[Task 2]	0.382	0.045	0.001	0.351	0.356	0.462	6001	320,000
ε	$8.657\times 10^{-7}$	$1.641\times 10^{-6}$	$4.859\times 10^{-8}$	$9.748\times 10^{-8}$	$2.294\times 10^{-7}$	$6.956\times 10^{-6}$	6001	320,000

**Table 3 healthcare-10-00226-t003:** Regression results for six factors.

Estimated Parameter	$\theta_{0}$	Gender	Job Title	Funding Status	LTCF Type	Facility Scale	Strategy Type
α	0.9	0.14	−0.27	−0.03	−0.06 *	0.1 *	−0.03 **
β	0.9	0.09 *	−0.16 **	−0.27 *	−0.03 *	0.05 *	0.14 **

Note: * *p*-value < 0.1, ** *p*-value < 0.01. Gender: female (coded as 1 in the regression analysis) and male (coded as 2); Job title: facility (1) and healthcare (2); Funding status: public (1) and for profit (2); LTCF type: nursing homes (1), residential homes (2), and mixed LTCFs (3); Scale: less than 99 beds (1), 100~399 beds (2), and more than 400 beds (3); Strategy type: cost-leadership (1), differentiation (2), and hybrid (3).

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
