# Peer review of "Perception Bias Effects on Healthcare Management in COVID-19 Pandemic: An Application of Cumulative Prospect Theory"

_healthcare, 2022, doi:10.3390/healthcare10020226_

Round 1

Reviewer 1 Report

It is a well written paper. It addresses perception biases of long-term care facilities (LTCFs) of managers. Paper presented three different methods.

  • Update address literature review on uncertainty in expert opinion/judgment by methods such as Dempster Shafer theory and evidential reasoning.
  • Type of questions in the survey. How it was designed to cover biases in LTCFs decision-makers?
  • What type of optimizer technique “Solver of OpenBugs 3.2.3” use? Give reference? Is this solver open source?

Author Response

Dear reviewer:

Please see the attached file. Thank you.

Reviewer 2 Report

Generally, the manuscript was a good presented. I have just few comments on data or data presentation that you can find below:

  1. Figure2: you need to add the horizontal and vertical label.
  2. You have selected six features(socidemogrphic) in your study. It is to add a verification for this selection.
  3. The survey was based on members of Taiwan Long- Term Care Profession Association. Is this considered as one Facility? If not just clarify the Facility scale type in your Table1.
  4. The conclusion  includes a lot of details. It is good to make short.

Author Response

(The authors gave the same response as above.)
